# Murine *Surf4* is essential for early embryonic development

**Brian T. Emmer**[1,2], **Paul J. Lascuna**[2], **Vi T. Tang**[2,3], **Emilee N. Kotnik**[2¤], **Thomas L. Saunders**[1,4,5], **Rami Khoriaty**[1,5,6,7], **David Ginsburg**[1,2,5,6,8,9,10]*

**1** Department of Internal Medicine, University of Michigan, Ann Arbor, Michigan, **2** Life Sciences Institute, University of Michigan, Ann Arbor, Michigan, **3** Department of Molecular and Integrative Physiology, University of Michigan, Ann Arbor, Michigan, **4** Transgenic Animal Model Core Laboratory, University of Michigan, Ann Arbor, Michigan, **5** University of Michigan Rogel Cancer Center, Ann Arbor, Michigan, **6** Cellular and Molecular Biology Program, University of Michigan, Ann Arbor, Michigan, **7** Department of Cell and Developmental Biology, University of Michigan, Ann Arbor, Michigan, **8** Department of Human Genetics, University of Michigan, Ann Arbor, Michigan, **9** Department of Pediatrics, University of Michigan, Ann Arbor, Michigan, **10** Howard Hughes Medical Institute, University of Michigan, Ann Arbor, Michigan

¤ Current address: Molecular Genetics and Genomics Program, Washington University in St. Louis, St. Louis, Missouri
* ginsburg@umich.edu

**Data Availability Statement:** All relevant data are within the paper and its Supporting Information files.

**Funding:** This research was supported by NIH grants R35-HL135793T (DG), T32-HL007853, KL2-TR002241 and K08-HL148552 (BTE). The

## Abstract

Newly synthesized proteins co-translationally inserted into the endoplasmic reticulum (ER) lumen may be recruited into anterograde transport vesicles by their association with specific cargo receptors. We recently identified a role for the cargo receptor SURF4 in facilitating the secretion of PCSK9 in cultured cells. To examine the function of SURF4 *in vivo*, we used CRISPR/Cas9-mediated gene editing to generate mice with germline loss-of-function mutations in *Surf4*. Heterozygous *Surf4*$^{+/-}$ mice exhibit grossly normal appearance, behavior, body weight, fecundity, and organ development, with no significant alterations in circulating plasma levels of PCSK9, apolipoprotein B, or total cholesterol, and a detectable accumulation of intrahepatic apoliprotein B. Homozygous *Surf4*$^{-/-}$ mice exhibit embryonic lethality, with complete loss of all *Surf4*$^{-/-}$ offspring between embryonic days 3.5 and 9.5. In contrast to the milder murine phenotypes associated with deficiency of known SURF4 cargoes, the embryonic lethality of *Surf4*$^{-/-}$ mice implies the existence of additional SURF4 cargoes or functions that are essential for murine early embryonic development.

## Introduction

The coatomer protein complex II (COPII) coat assembles on the cytoplasmic surface of endoplasmic reticulum (ER) exit sites to drive the formation of membrane-bound transport vesicles. Efficient recruitment of proteins and lipids into these vesicles occurs via physical interaction with the COPII coat[1]. For cargoes accessible on the cytoplasmic surface of the ER membrane, this interaction may be direct. For soluble cargoes in the ER lumen, however, transmembrane cargo receptors serve as intermediaries for this interaction[2]. Although thousands of human proteins traffic through the secretory pathway, a corresponding cargo receptor

funders had no role in study design, data collection and analysis, decision to publish, or preparation of the manuscript.

**Competing interests:** DG is a Howard Hughes Medical Institute investigator. This does not alter our adherence to PLOS ONE policies on sharing data and materials.

has been identified for only a few, and the size and identity of the cargo repertoire for each individual cargo receptor remains largely unknown.

Through unbiased genome-scale CRISPR screening, we recently discovered a role for the ER cargo receptor Surfeit locus protein 4 (SURF4) in the secretion of Proprotein convertase subtilisin/kexin type 9 (PCSK9)[3], a protein that modulates mammalian cholesterol homeostasis through its negative regulation of the Low-density lipoprotein receptor (LDLR)[4]. Consistent with a role as a PCSK9 cargo receptor, SURF4 was found to localize to the ER and the ER-Golgi intermediate compartment, to physically associate with PCSK9, and to promote the ER exit and extracellular secretion of PCSK9. These experiments relied on heterologous expression of PCSK9 in HEK293T cells, however, and the physiologic relevance of this interaction *in vivo* remains uncertain. Additionally, although SURF4 deletion did not affect the secretion of a control protein, alpha-1 antitrypsin, a broader role for SURF4 in protein secretion remains possible and is supported by the recent identification of other potential cargoes including apolipoprotein B, growth hormone, dentin sialophosphoprotein, and amelogenin[5, 6].

To investigate the physiologic functions of SURF4, we generated mice with targeted disruption of the *Surf4* gene. We found that partial loss of SURF4 in heterozygous mice led to a modest accumulation of intrahepatic apolipoprotein B, with no effect on steady state plasma levels. However, complete genetic deletion of *Surf4* resulted in early embryonic lethality.

## Results

### Generation of mice with germline deletion of *Surf4*

The *Mus musculus Surf4* gene is composed of 6 exons and 5 introns spanning approximately 14 kb in the tightly clustered surfeit gene locus on chromosome 2[7, 8]. We targeted exon 2 of *Surf4* for CRISPR/Cas-mediated mutagenesis (Fig 1A), verified sgRNA efficiency in embryonic stem (ES) cells (Fig 1B and 1C), and generated mice from microinjected zygotes. Sanger sequencing identified 4 of 57 mice with disruption of the target site (Fig 1D). These mice were then mated to C57BL/6J wild-type mice and their progeny genotyped, confirming germline transmission for each of the 4 alleles (Fig 1E). Two alleles introduced frameshift deletions both leading to early termination codons, with the other alleles containing in-frame deletions of 3 and 6 DNA base pairs, respectively.

### Effect of SURF4 haploinsufficiency on cholesterol regulation

*Surf4*$^{+/-}$ mice were observed at expected Mendelian ratios at weaning (Table 1) and exhibited grossly normal appearance, behavior, and organ development by necropsy. Analysis of mRNA from *Surf4*$^{+/-}$ mouse liver tissue confirmed a reduction in total *Surf4* transcripts with a relative decrease in the mutant allele, consistent with nonsense-mediated decay (Fig 2). *Surf4*$^{+/-}$ mice heterozygous for the del(1) allele also showed no significant differences in plasma PCSK9, cholesterol, and apolipoprotein B levels compared to *Surf4*$^{+/+}$ litter-mate controls (Fig 3). Similarly, no differences were seen in the intrahepatic accumulation of the putative SURF4 cargo, PCSK9, or its downstream target, LDL receptor. In contrast, despite its normal steady state levels in plasma, an approximately 3-fold increase in intrahepatic apolipoprotein B was observed in *Surf4*$^{+/-}$ mice (Fig 4), consistent with selective retention of this putative SURF4 cargo in the setting of *Surf4* haploinsufficiency.

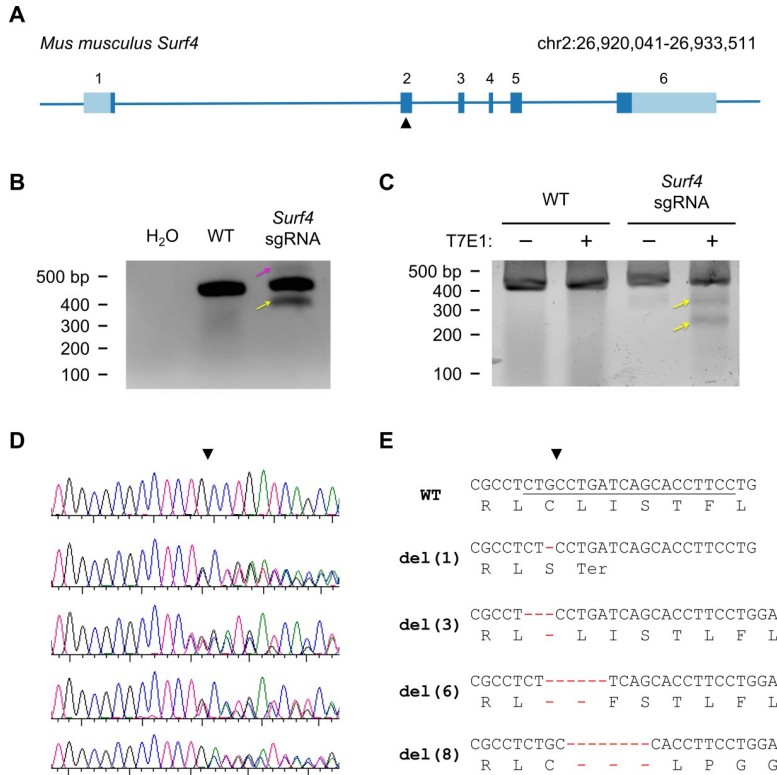

**Fig 1. Generation of *Surf4* mutant alleles.** (A) *Surf4* gene structure. Exons are shaded light blue for untranslated regions or dark blue for coding sequence. The target site for the sgRNA used for oocyte editing is indicated by the black triangle. (B) Mouse ES cells were either untreated or electroporated with plasmids for CRISPR/Cas9 disruption of the *Surf4* target site. PCR amplification of genomic DNA or water control across the *Surf4* target site revealed higher and lower molecular weight DNA fragments suggestive of nonhomologous endjoining repair of Surf4 indels. (C) The major PCR product was gel purified and subjected to T7 endonuclease I digestion. T7E1 digestion produced novel DNA fragments (arrows) indicating the presence of insertions/deletions in *Surf4* exon 2. Wild type DNA was resistant to T7E1 digestion. (D) Sanger sequencing chromatograms of *Surf4* target site amplicons of progeny from matings between *Surf4*-targeted founder mice and wild-type C57BL6/J mice. (E) DNA and predicted protein sequences for the 4 individual allele generated by CRISPR/Cas9 gene-editing of *Surf4*.

## *Surf4* function is required for embryonic development

Intercrosses were performed for *Surf4*[+/-] mice carrying each of the 3 independent *Surf4* deletion alleles described above. Genotyping at the time of weaning demonstrated the expected number of heterozygous progeny, with complete absence of homozygous *Surf4*[-/-] pups (Table 2). Timed matings of mice heterozygous for the del(1) allele were performed, with no *Surf4*[-/-] embryos identified at E9.5 or later (Table 3). However, analysis of E3.5 blastocysts

**Table 1. Mice heterozygous for each of 3 independent *Surf4* targeted alleles are observed at expected Mendelian ratios at weaning.** Mice heterozygous for the indicated *Surf4* allele were crossed with wild-type C57BL/6J mice and the resulting litters genotyped for the corresponding *Surf4* alleles at approximately 2 weeks of age. The proportion of mice with the heterozygous mutant genotype was compared to expected Mendelian ratios by the chi-square test.

| Allele | *Surf4*[+/-] x *Surf4*[+/+] | | $p$ ($X^2$) |
|---|---|---|---|
| | Progeny | | |
| | +/+ | +/- | |
| del (3) | 36 | 39 | 0.81 |
| del (6) | 22 | 15 | 0.41 |
| del (1) | 19 | 18 | 0.91 |

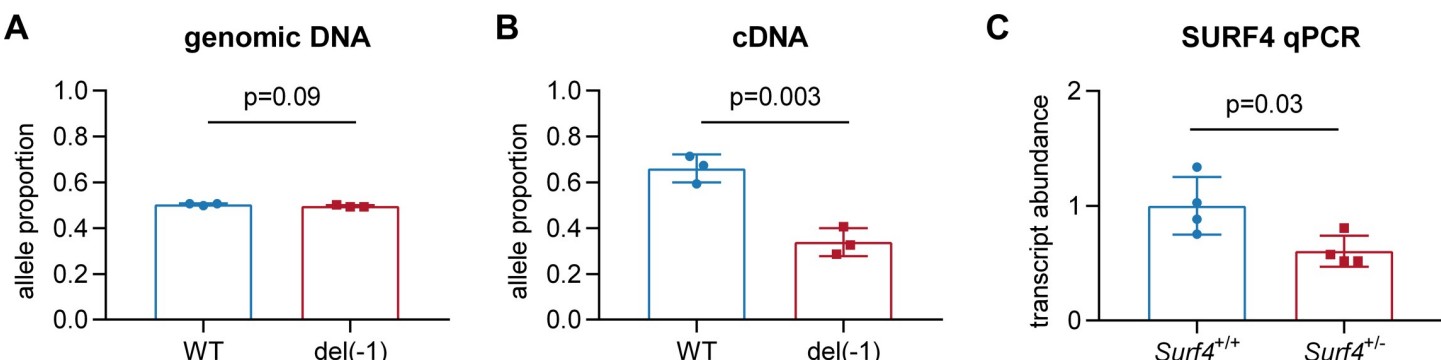

**Fig 2. *Surf4^+/-* mice exhibit partial reduction of *Surf4* transcripts with preferential loss of the mutant allele.** Sanger sequencing of the *Surf4* target site was performed on PCR amplicons derived from genomic DNA (A) or reverse-transcribed cDNA (B) prepared from liver tissue of 3 *Surf4^+/-* mice. Decomposition of chromatograms was performed to quantify the relative proportion of each allele in each sample. (C) Total *Surf4* transcript levels in liver tissue from 4 *Surf4^+/+* and 4 *Surf4^+/-* mice were quantified and normalized to a panel of housekeeping genes by qRT-PCR.

generated by *in vitro* fertilization revealed the expected proportion of *Surf4^-/-* genotypes with no gross morphologic abnormalities on visual assessment by an experienced expert in murine embryology Thus, complete genetic deficiency of *Surf4* results in embryonic lethality occurring sometime between E3.5 and E9.5.

## Discussion

Identification of the molecular machinery underlying eukaryotic protein secretion has been elucidated by elegant work in model systems including yeast and cultured mammalian cells. Recent characterizations of mice with genetic deficiency of COPII components have extended these findings to mammalian physiology, revealing a variety of complex phenotypes[9–18]. Comparatively little is known about the physiologic role of mammalian cargo receptors *in vivo*. In humans, genetic deletion of either subunit of a cargo receptor complex, LMAN1/ MCFD2, results in a rare bleeding disorder due to the impaired secretion of coagulation factors V and VIII[19, 20], with a similar phenotype in *Lman1^-/-* and *MCFD2^-/-* mice[21, 22].

We set out to investigate the physiologic function of murine SURF4 *in vivo*, with a particular focus on its putative function in the secretion of PCSK9[3] and apolipoprotein B[6], both of which play central roles in mammalian cholesterol regulation. We generated multiple independent gene targeted *Surf4* alleles, with heterozygous *Surf4^+/-* mice exhibiting no gross

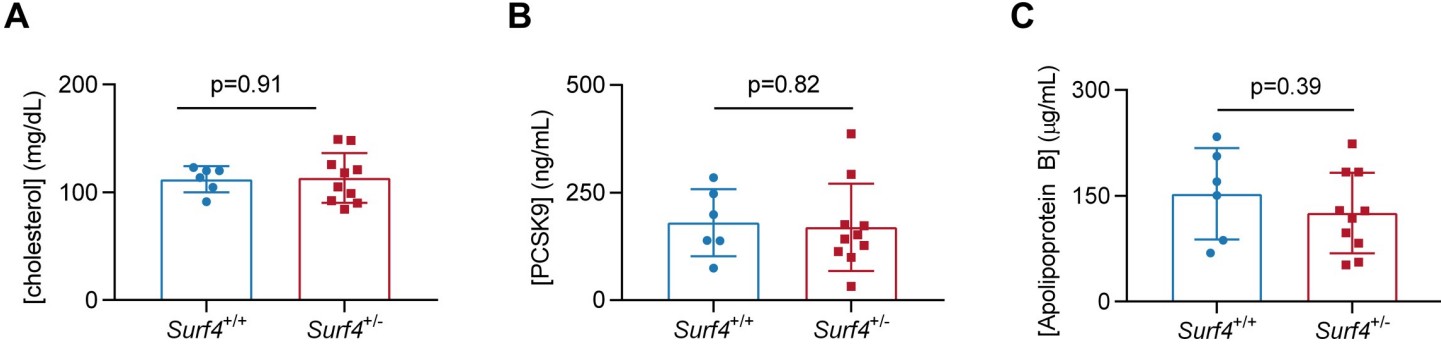

**Fig 3. *Surf4* haploinsufficiency does not affect baseline plasma levels for PCSK9, ApoB, or cholesterol levels.** Plasma samples collected from 10 *Surf4^+/-* mice (heterozygous for the del(1) allele) and 6 wild-type littermate controls were assayed for plasma levels of total cholesterol (A), PCSK9 (B), and ApoB (C). Values were measured and averaged for each of two independent phlebotomies from each mouse. Both male and female mice were tested for each genotype. Significance testing was calculated by Student's t-test between genotype groups.

## A

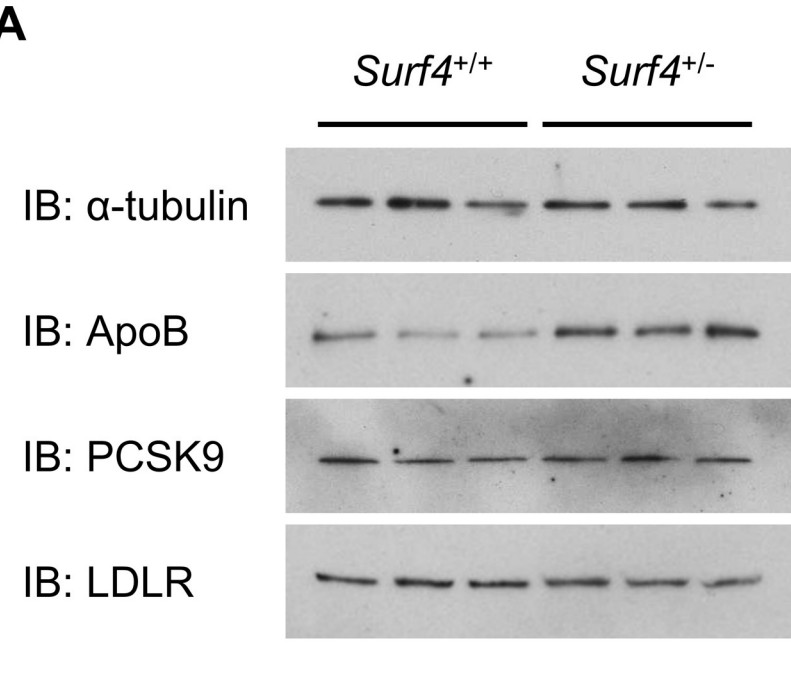

## B

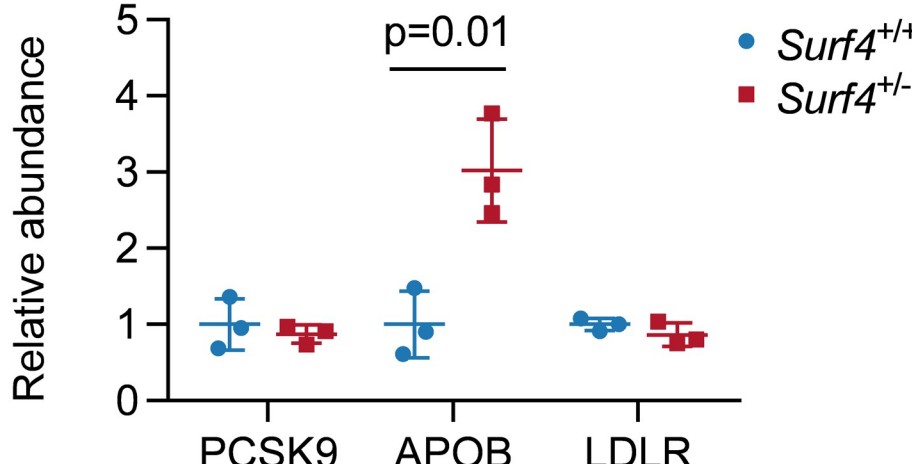

**Fig 4. Surf4 haploinsufficiency causes hepatic accumulation of apolipoprotein B but not PCSK9.** Liver lysates from 3 male *Surf4*+/- mice harboring the del(1) allele and 3 male *Surf4*+/+ littermate controls were immunoblotted for PCSK9, LDL receptor, apolipoprotein B, and alpha-tubulin. Densitometry values for PCSK9, LDLR, and apolipoprotein B were normalized to alpha-tubulin. Significance testing was performed by Student's t-test between genotype groups.

developmental abnormalities and normal circulating levels of cholesterol, PCSK9, and apolipoprotein B. Consistent with our observations that SURF4 haploinsufficiency is well-tolerated in mice, a number of loss-of-function variants have been observed in human *SURF4*, including a p.Gln185Ter nonsense variant with an allele frequency of 0.1%[23]. Of note, previous human

**Table 2. Germline deletion of *Surf4* causes embryonic lethality.** Mice heterozygous for the indicated *Surf4* alleles were intercrossed and progeny genotyped for *Surf4* at weaning. The proportion of mice with the homozygous null genotype was compared to expected Mendelian ratios by the chi-square test.

| | | *Surf4*⁺/⁻ x *Surf4*⁺/⁻ | | | |
|---|---|---|---|---|---|
| **Allele** | **Stage** | **Progeny** | | | **$p$ ($X^2$)** |
| | | **+/+** | **+/-** | **-/-** | |
| del (3) | weaning | 9 | 22 | 0 | <0.01 |
| del (6) | weaning | 12 | 32 | 0 | <0.01 |
| del (1) | weaning | 40 | 72 | 0 | <0.01 |

genome-wide association studies for lipid traits have not detected a significant signal near the *SURF4* gene[24].

To assess the impact of *Surf4* haploinsufficiency on the physiologic secretion of putative cargoes, we measured the levels of PCSK9 and apolipoprotein B in circulation and in the liver.

We found that plasma and intrahepatic levels of PCSK9 were unaffected by partial SURF4 reduction in *Surf4*⁺/⁻ mice. The *Surf4* exon targeted by our gene editing approach is expressed in all currently annotated mouse *Surf4* splice variants (Ensembl release 98[25]). Analysis of liver mRNA from *Surf4*⁺/⁻ mice confirmed a reduction in total *Surf4* transcript levels, with reduced levels of the mutant allele relative to the wild-type allele consistent with nonsense-mediated decay (Fig 2). This observation is similar to the normal plasma levels of LMAN1 cargo proteins reported in heterozygous *Lman1*⁺/⁻ mice[21]. In contrast, we found that apolipoprotein B accumulated approximately 3-fold in liver cell lysates prepared from *Surf4*⁺/⁻ mice compared to controls, suggesting greater sensitivity of apolipoprotein B than PCSK9 to partial SURF4 depletion. Nonetheless, plasma levels of apolipoprotein B and total cholesterol were unaffected by haploinsufficiency for *Surf4*, possibly due to downstream effects on complex cholesterol regulatory pathways which could alter the rate of clearance and/or expression of ApoB or other related components of this network. Together, these observations indicate a complexity in the degree of dependence of different cargoes on the partial or complete reduction of their corresponding cargo receptor. The mechanistic basis for this variability remains unknown but may be related to different stoichiometries or cargo receptor binding affinities.

In cultured cells, secretion defects of PCSK9[3] and apolipoprotein B[6] are observed upon complete deletion of *SURF4*. Our attempts to generate adult mice with complete loss of *Surf4* were precluded by the embryonic lethality caused by *Surf4* deletion. A loss of *Surf4*⁻/⁻ mice at weaning was unlikely to have been caused by a linked spontaneous or off-target CRISPR-generated passenger mutation[26] as this phenotype was observed for each of 3 independent alleles. Timed matings revealed that loss of *Surf4*⁻/⁻ embryos occurs between E3.5 and E9.5.

**Table 3. Germline deletion of Surf4 results in lethality between embryonic day 3.5 and 9.5.** Timed matings were performed between *Surf4*⁺/⁻ mice carrying the del(1) allele and embryos harvested at E9.5, E12.5, E.14.5 or at the time of weaning. For analysis at E3.5, blastocysts were collected following *in vitro* fertilization of oocytes from *Surf4*⁺/⁻ females with sperm from *Surf4*⁺/⁻ males. The proportion of mice with the homozygous null genotype was compared to expected Mendelian ratios by the chi-square test.

| | | *Surf4*⁺/⁻ x *Surf4*⁺/⁻ | | | |
|---|---|---|---|---|---|
| **Allele** | **Stage** | **Progeny** | | | **$p$ ($X^2$)** |
| | | **+/+** | **+/-** | **-/-** | |
| del (1) | weaning | 40 | 72 | 0 | <0.01 |
| | E14.5 | 2 | 3 | 0 | >0.99 |
| | E12.5 | 3 | 12 | 0 | 0.100 |
| | E9.5 | 5 | 15 | 0 | 0.047 |
| | E3.5 | 1 | 7 | 4 | >0.99 |

The mechanism for this observation is unclear. Deficiency of the SURF4 homologue SFT-4 is similarly associated with embryonic lethality in *C. elegans*[6], but SURF4 is not essential for cellular viability in cultured HEK293T cells[3, 5] and its homologue, Erv29p, is dispensable in yeast[27, 28]. Deficiencies of PCSK9 or apoliporotein B alone cannot account for this developmental phenotype, given that PCSK9$^{-/-}$ mice are viable[29] and that ApoB$^{-/-}$ mice survive past E9.5[30]. Likewise, mice with genetic deletion of 3 other putative SURF4 cargoes, growth hormone[31], amelogenin[32], and dental sialophosphoprotein[33], are viable. The embryonic lethality of SURF4 deficiency may therefore result from additive effects of disrupted secretion of known cargoes, or the presence of additional unknown SURF4 cargoes or functions that are essential for early embryonic development. A role for SURF4 in global protein secretion is unlikely, as previous studies demonstrated no effect of SURF4 deletion on the secretion of a number of other proteins[3, 5]. A broader role for SURF4 in the secretion of additional unknown cargoes however is suggested by the observation that SURF4 has been evolutionarily conserved in yeast and other organisms lacking homologues of PCSK9. An N-terminal tripeptide "ER-ESCAPE motif", present on a large number of potential cargoes, has recently been proposed to mediate cargo recruitment by SURF4[5]. A comprehensive identification of SURF4 cargoes and the nature of their interaction with SURF4 should clarify the function of SURF4 in cholesterol regulation and in mammalian development.

## Materials and methods

### Generation of *Surf4* mutant mice

All animal protocols used in this study were approved by the University of Michigan Committee on the Use and Care of Animals. We used CRISPR/Cas9 technology[34, 35] to generate a new genetically modified mouse strain with a *Surf4* gene knockout. The presence of a premature termination codon in exon 2 is predicted to result in loss of protein expression due to nonsense mediated decay of mRNA[36]. A single guide RNA (sgRNA) target and protospacer adjacent motif was identified in exon 2 (ENSMUSE00000232711.1) with CRISPOR[37]. The sgRNA is 5′ CTGCCTGATCAGCACCTTCC TGG 3′ on the non-coding strand (chromosome 2; coordinates 26926892–26926911) with a predicted cut site 47 bp downstream of the first exon 2 codon. The sgRNA target was cloned into plasmid pX330 (Addgene.org plasmid #42230, a kind gift of Feng Zhang) as described[38]. The sgRNA was validated in mouse JM8. A3 ES cells[39] prior to use for mouse zygote microinjection. The sgRNA plasmid (15 μg) was electroporated into 8 X 10E6 ES cells. To the electroporation, 5 μg of a PGK1-puromycin resistance plasmid[40] was added for transient puromycin selection (2 μg/ml puromycin applied 48–72 hours after electroporation). ES cell culture and electroporation was carried out as described[41]. After selection, DNA was extracted from surviving cells, PCR was used to amplify the sequences across the sgRNA cut site, and T7 endonuclease 1 assays were used to detect small deletions/insertions at the predicted Cas9 DNA cut site[42]. The circular sgRNA plasmid was resuspended in microinjection buffer as described[43]. The plasmid mixture was used for pronuclear microinjection of zygotes obtained from the mating of superovulated C57BL/6J female mice (The Jackson Laboratory Stock No. 0006640) and C57BL/6J male mice as described[44]. A total of 305 zygotes were microinjected and 285 zygotes were transferred to pseudopregnant B6D2F1 female mice (The Jackson Laboratory Stock No. 100006). 18 mouse pups were born and four of them transmitted gene edited *Surf4* alleles.

### Mouse genotyping

*Surf4* genotyping was performed by PCR of genomic DNA with primers mSurf4-ex2-for [TGCTGAGGGCCTCTCTGTCT] and mSurf4-ex2-rev [CAGGTAGCCACAGCTCCAGG]. Sanger

sequencing was performed with the same genotyping primers and chromatograms were inspected both manually and by automated deconvolution[45] to determine the presence or absence of target site indels.

## Analysis of *Surf4*⁺/⁻ mice

Mice were housed and monitored in accordance with University of Michigan Unit of Laboratory Animal Medicine (ULAM) guidelines. Blood was collected at 6–12 weeks of age by retro-orbital bleeding into heparin-coated collection tubes from mice anesthetized with isoflurane. Plasma was prepared by centrifugation at 2,000 g for 10 min at 4˚C. A second blood collection was performed 1 week following the initial collection. Plasma samples were analyzed by total cholesterol colorimetric assay (SB-1010-225, Fisher Scientific, Hampton NH) and ELISAs for PCSK9 (MPC900, R&D Systems, Minneapolis MN) and apolipoprotein B (ab230932, Abcam, Cambridge UK). Liver tissue was perfused with PBS and harvested from mice at the time of sacrifice. Liver mRNA was prepared with RNeasy Plus Mini Kit (Qiagen, Hilden, Germany) and oligo(dT)-primed first strand cDNA generated with Superscript III reverse transcriptase (Invitrogen, Carlsbad CA) according to manufacturer's instructions. Quantitative PCR was performed using 20 ng of cDNA per reaction with PowerSYBR Green PCR Master Mix (Applied Biosystems, Foster City CA). Normalization of qPCR data was performed using a panel of 4 selected housekeeping genes[46]. The relative proportion of each *Surf4* allele was quantified by decomposition of Sanger sequencing chromatograms with TIDE indel analysis [45]. The primer sequences for qRT-PCR were: *Surf4*-forward [CTGTTGGCCTCATCCTTCGT], *Surf4*-reverse [GGCAATTGTCTGCAGTGCG], *Actb*-forward [CCACTGCCGCATCCTCTTCC], *Actb*-reverse [CTCGTTGCCAATAGTGATGACCTG], *B2m*-forward [CATGGCTCGCTCGGTGACC], *B2m*-reverse [AATGTGAGGCGGGTGGAACTG], *Tbp*-forward [CCCCACAACTCTTCCATTCT], *Tbp*-reverse [GCAGGAGTGATAGGGGTCAT], *Ppia*-forward [CAAATGCTGGACCAAACACAAACG], *Ppia*-reverse [GTTCATGCCTTCTTTCACCTTCCC].

Protein lysates were prepared from liver tissue by mechanical homogenization, resuspension in RIPA lysis buffer (Pierce Manufacturing, Appleton WI), and collection of supernatant after centrifugation for 15 minutes at 21,000x*g*. Protein concentrations of lysates were measured by DC Protein Assay (5000111, Bio-Rad Laboratories, Hercules CA). Equal amounts of lysate were analyzed by immunoblotting with antibodies against Apolipoprotein B (70R-15771, 1:1000, Fitzgerald Industries International, Acton MA), PCSK9 (ab31762, 1:1000, Abcam), LDLR (ab52818, 1:1000, Abcam), and alpha-tubulin (ab176560, 1:2000, Abcam) and densitometry analysis performed with ImageJ[47].

## Timed matings and *in vitro* fertilization

For analysis of embryonic day 9.5 and later, *Surf4*⁺/⁻ male and female mice were co-housed overnight and females with copulatory vaginal plugs the following morning were assigned embryonic day 0.5. Pregnant females were then sacrificed at indicated time points and genomic DNA prepared from isolated embryos. For analysis of embryonic day 3.5, *Surf4*⁺/⁻ females were superovulated with anti-inhibin serum as described[48]. The collected oocytes were fertilized with sperm from *Surf4*⁺/⁻ males as described[49]. Resulting fertilized eggs were maintained in cell culture in KSOM medium (Zenith Biotech) until visual inspection, harvesting, and genomic DNA preparation from blastocysts at embryonic day 3.5.

## Author Contributions

**Conceptualization:** Brian T. Emmer, Thomas L. Saunders, David Ginsburg.

**Data curation:** Brian T. Emmer, Paul J. Lascuna, Vi T. Tang, Emilee N. Kotnik, Thomas L. Saunders, Rami Khoriaty.

**Formal analysis:** Brian T. Emmer, David Ginsburg.

**Funding acquisition:** David Ginsburg.

**Methodology:** Brian T. Emmer.

**Writing – original draft:** Brian T. Emmer.

**Writing – review & editing:** Brian T. Emmer, Thomas L. Saunders, Rami Khoriaty, David Ginsburg.

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
