## [Decision Letter · Decision Letter 0]

22 Aug 2019

PONE-D-19-17413

Murine Surf4 is essential for early embryonic development

PLOS ONE

Dear Dr. Emmer,

Thank you for submitting your manuscript to PLOS ONE. After careful consideration, we feel that it has merit but does not fully meet PLOS ONE’s publication criteria as it currently stands. Therefore, we invite you to submit a revised version of the manuscript that addresses the points raised during the review process.

As you will read, both reviewer point the same critique: to show the effect of the heterozygous condition of Surf4 at the protein level to support your analysis. It is not clear to me if you have observed this early homozygote lethality with another clone of Surf4 inactivation or in another genetic background.It would be nice to indicate this information clearly.

We would appreciate receiving your revised manuscript by Oct 06 2019 11:59PM. To enhance the reproducibility of your results, we recommend that if applicable you deposit your laboratory protocols in protocols.io, where a protocol can be assigned its own identifier (DOI) such that it can be cited independently in the future. For instructions see: http://journals.plos.org/plosone/s/submission-guidelines#loc-laboratory-protocols

We look forward to receiving your revised manuscript.

Kind regards,

Yann Herault

Academic Editor

PLOS ONE

Journal Requirements:

1. Please include your tables as part of your main manuscript and remove the individual files. Please note that supplementary tables (should remain/ be uploaded) as separate "supporting information" files

Reviewers' comments:

Reviewer's Responses to Questions

**Comments to the Author**

1. Is the manuscript technically sound, and do the data support the conclusions?

Reviewer #1: Partly

Reviewer #2: Partly

2. Has the statistical analysis been performed appropriately and rigorously? 

Reviewer #1: Yes

Reviewer #2: Yes

3. Have the authors made all data underlying the findings in their manuscript fully available?

Reviewer #1: Yes

Reviewer #2: Yes

4. Is the manuscript presented in an intelligible fashion and written in standard English?

Reviewer #1: Yes

Reviewer #2: Yes

5. Review Comments to the Author

Reviewer #1: PLoS One Review of Murine Surf4 is essential for early embryonic development (Emmer et al)

In this manuscript the authors generate a null allele of Surf2, an ER cargo receptor and one they previously identified as interacting with PCSK9 and thus one that may have important roles in cholesterol homeostasis. Overexpression and deletion of Surf2 in cell lines were supportive of this role and they sought to determine if this role was recapitulated in the mouse. They generated 4 lines via a Crispr strategy including 4 putative null lines. Heterozygous animals were generated at expected frequencies but null mice were early embryonic lethal. They find that hets have no changes in plasma levels of cholesterol, PCSK9 or ApoB but do observe significant changes in the abundance of ApoB in the livers of het mice.

This paper represents a natural extension of their previous findings and should be of interest to the same audience.

Several Major issues include:

1) The authors do not present evidence SURF4 is reduced in hets. They do, however repeatedly state that SURF4 (protein) is reduced in heterozygous animals. This information is critical to support the conclusions made herein.

2) To make definitive conclusions, the heterozygote data presented with a single line should be performed with more than one of the founder lines. The sex and number of animals used in Figs 2 and 3 should also be provided.

3) No methodology is provide for the data presented in Figure 3. The authors state that liver cell lysates were produced in the text but no information on the antibodies or assays used were provided. It would be important to include a blot demonstrating a reduction in SURF4 in Hets in this figure.

4) Images of the mutant blastocysts that were created via IVF should be provided in order for experts to decide if there are obvious defects.

Other

Abbreviations should be named prior to use: SURF2, ERGIC, LDLR, PCSK9 etc are never defined.

Reviewer #2: The manuscript “Murine Surf4 is essential for early embryonic development” by Emmer et al describes the generation of a new mouse knockout model for the anterograde cargo receptor SURF4. The authors used CRISPR/Cas-mediated mutagenesis to target exon2 of the Surf4 gene, generating 4 gene-edited lines (Fig. 1). They then analyzed one of those lines in heterozygous adult mice, by assessing plasma samples for total cholesterol, PCSK9, and ApoB (Fig. 2). PCSK9 and ApoB are two candidate proteins that are predicted to be trafficked by SURF4 from the ER to Golgi based on published literature, and total cholesterol could have been impacted if PCSK9 and ApoB were not synthesized and secreted properly by SURF4-mutant cells. Seeing no effects on Surf4+/- plasma, they analyzed total liver lysates for ApoB, PCSK9, and LDLR (which is regulated by PCSK9) by western blotting (Fig. 3). They found an aberrant accumulation of ApoB in the liver lysates, although this did not affect circulating ApoB levels, as demonstrated by the plasma ELISAs. The authors could not analyze Surf4-/- mice because they were not recovered at weaning from three of their mutant lines (Table 2). They analyzed one of the lines with timed matings to assess when the Surf4-/- embryos died but could not recover live or partially resorbed Surf4-/- embryos at E9.5—the earliest timepoint they analyzed. They also performed in vitro fertilization with Surf4+/- oocytes and sperm and cultured fertilized eggs for 3 days until the blastocyst stage. This approach did generate Surf4-/- blastocysts, so the authors concluded that Surf4-/- embryos died en utero between E3.5 and E9.5 from unknown causes.

This manuscript provides the first description of a Surf4-deficient mouse, which is a critical reagent for interpreting the in vitro data that has been generated on SURF4 and its function in different cell types. The authors particularly hoped to validate that the PCSK9 protein—which they had described as being trafficked by SURF4 in vitro (Emmer et al, Elife, 2018)—would also be found to be trafficked in vivo, with possible implications for LDLR surface expression and plasma cholesterol levels. However, the Surf4+/- mice they analyzed did not support a critical role for SURF4 in this capacity. They did, however, find accumulation of ApoB in the livers of Surf4+/- mice, which supports published evidence that SURF4 helps to traffic ApoB for proper secretion from hepatocytes (Saegusa et al, J Cell Biol, 2018). Curiously, this did not alter circulating ApoB plasma levels or cholesterol levels, which the authors could not explain. Altogether, the conclusion from these analyses is that Surf4 haploinsufficiency does not impact total cholesterol levels in circulation.

While I appreciate the importance of the new Surf4-mutant lines that this manuscript describes, I would have liked to have seen a better analysis of those lines. Specifically, I wish the authors had made some effort to confirm the reduction of SURF4 in their heterozygous adult mice—either by immunoblotting or qPCR. The Surf4-/- blastocysts they generated also could have been analyzed by qPCR. I believe that more careful validation of the mutant lines and of Surf4 reduction is important because Surf4 is known to be subject to alternative splicing (Garson et al, Gene Expr, 1996). In addition, I found the developmental analysis of Surf4-/- embryos to be perfunctory. The fact that no partially resorbed Surf4-/- embryos could be recovered at E9.5 suggests that the embryos die much earlier—potentially around implantation. I found the in vitro generation of Surf4-/- blastocysts to be an interesting approach to assessing survivability of Surf4-/- embryos at early stages of development, although the artificiality of this approach could be misleading in interpreting early Surf4-/- embryonic development. Instead, I would have preferred to see the authors flush blastocysts from pregnant mice at E3.5 to assess the feasibility of early Surf4-/- development. Furthermore, analysis of embryos at peri-implantation (E5.5-6.5) would be a significant improvement to this study, since it would clarify whether Surf4-/- embryos can implant. Methods for dissecting peri-implantation mouse embryos are well described (Shea and Geijsen, JoVE, 2007). Altogether, better analysis of the timepoint at which Surf4-/- embryos die would assist in future studies that will elucidate SURF4 cargo proteins that are critical for early development.

6. PLOS authors have the option to publish the peer review history of their article (what does this mean?). If published, this will include your full peer review and any attached files.

Reviewer #1: No

Reviewer #2: No

---

## [Author Response · Author response to Decision Letter 0]

14 Nov 2019

A full point-by-point response to editor and reviewer comments has been uploaded as a separate file.

---

## [Decision Letter · Decision Letter 1]

19 Dec 2019

Murine Surf4 is essential for early embryonic development

PONE-D-19-17413R1

Dear Dr. Emmer,

We are pleased to inform you that your manuscript has been judged scientifically suitable for publication and will be formally accepted for publication once it complies with all outstanding technical requirements.

With kind regards,

Yann Herault

Academic Editor

PLOS ONE

Additional Editor Comments (optional):

Please be sure to include all the methodology and technical details needed for the publication.

Reviewers' comments:

Reviewer's Responses to Questions

**Comments to the Author**

1. If the authors have adequately addressed your comments raised in a previous round of review and you feel that this manuscript is now acceptable for publication, you may indicate that here to bypass the “Comments to the Author” section, enter your conflict of interest statement in the “Confidential to Editor” section, and submit your "Accept" recommendation.

Reviewer #1: (No Response)

Reviewer #2: All comments have been addressed

2. Is the manuscript technically sound, and do the data support the conclusions?

Reviewer #1: Yes

Reviewer #2: Yes

3. Has the statistical analysis been performed appropriately and rigorously? 

Reviewer #1: Yes

Reviewer #2: Yes

4. Have the authors made all data underlying the findings in their manuscript fully available?

Reviewer #1: Yes

Reviewer #2: Yes

5. Is the manuscript presented in an intelligible fashion and written in standard English?

Reviewer #1: Yes

Reviewer #2: Yes

6. Review Comments to the Author

Reviewer #1: The authors have adequately revised the MS. Given the amount of key information that was missing from the initial submission I suggest that the authors make sure that they are providing all key methodology and reagents for the final paper.

Reviewer #2: The revised manuscript now confirms reduction of transcripts in heterozygous mice and embryonic lethality in 3 separate homozygous lines. Although the timing and cause of the embryonic lethality is still unclear, the manuscript represents a sufficient initial description of the importance of SURF4 for embryonic development. Subsequent studies with a floxed Surf4 allele will presumably clarify the cargo and role of SURF4 in development and in specific cell types.

7. PLOS authors have the option to publish the peer review history of their article (what does this mean?). If published, this will include your full peer review and any attached files.

Reviewer #1: No

Reviewer #2: No

---

## [Editor Report · Acceptance letter]

14 Jan 2020

PONE-D-19-17413R1 

Murine *Surf4* is essential for early embryonic development 

Dear Dr. Emmer:

I am pleased to inform you that your manuscript has been deemed suitable for publication in PLOS ONE. Congratulations! Your manuscript is now with our production department. 

With kind regards,

on behalf of

Dr Yann Herault 

Academic Editor

PLOS ONE